# Serotonin Promotes Serum Albumin Interaction with the Monomeric Amyloid β Peptide

**DOI:** 10.3390/ijms22115896

**Published:** 2021-05-31

**Authors:** Ekaterina A. Litus, Alexey S. Kazakov, Evgenia I. Deryusheva, Ekaterina L. Nemashkalova, Marina P. Shevelyova, Aliya A. Nazipova, Maria E. Permyakova, Elena V. Raznikova, Vladimir N. Uversky, Sergei E. Permyakov

**Affiliations:** 1Institute for Biological Instrumentation, Pushchino Scientific Center for Biological Research of the Russian Academy of Sciences, Pushchino, 142290 Moscow, Russia; ealitus@gmail.com (E.A.L.); fenixfly@yandex.ru (A.S.K.); janed1986@ya.ru (E.I.D.); elnemashkalova@gmail.com (E.L.N.); marina.shevelyova@gmail.com (M.P.S.); alija-alex@rambler.ru (A.A.N.); mperm1977@gmail.com (M.E.P.); raznikova.elena@yandex.ru (E.V.R.); 2Department of Molecular Medicine and USF Health Byrd Alzheimer’s Research Institute, Morsani College of Medicine, University of South Florida, Tampa, FL 33612, USA

**Keywords:** Alzheimer’s disease, human serum albumin, amyloid β peptide, serotonin, tryptophan, surface plasmon resonance, molecular docking, intrinsic disorder

## Abstract

Prevention of amyloid β peptide (Aβ) deposition via facilitation of Aβ binding to its natural depot, human serum albumin (HSA), is a promising approach to preclude Alzheimer’s disease (AD) onset and progression. Previously, we demonstrated the ability of natural HSA ligands, fatty acids, to improve the affinity of this protein to monomeric Aβ by a factor of 3 (BBRC, 510(2), 248–253). Using plasmon resonance spectroscopy, we show here that another HSA ligand related to AD pathogenesis, serotonin (SRO), increases the affinity of the Aβ monomer to HSA by a factor of 7/17 for Aβ_40_/Aβ_42_, respectively. Meanwhile, the structurally homologous SRO precursor, tryptophan (TRP), does not affect HSA’s affinity to monomeric Aβ, despite slowdown of the association and dissociation processes. Crosslinking with glutaraldehyde and dynamic light scattering experiments reveal that, compared with the TRP-induced effects, SRO binding causes more marked changes in the quaternary structure of HSA. Furthermore, molecular docking reveals distinct structural differences between SRO/TRP complexes with HSA. The disintegration of the serotonergic system during AD pathogenesis may contribute to Aβ release from HSA in the central nervous system due to impairment of the SRO-mediated Aβ trapping by HSA.

## 1. Introduction

Human serum albumin (HSA; 66.5 kDa) is the most abundant blood and cerebrospinal fluid (CSF) protein and is vital to maintaining the osmotic pressure of blood and the transport of numerous substances, including fatty and amino acids, hormones, metal ions, and over 250 drugs (reviewed in [1,2]). HSA consists of three structurally similar α-helical domains comprising nine fatty-acid-binding sites, four metal-binding sites, and two high-affinity drug-binding sites [2]. One of the key participants in progression of Alzheimer’s disease (AD), amyloid peptide (Aβ) [3], is mostly bound to HSA (ca 89% of Aβ in blood plasma [4]). This interaction effectively rescues Aβ from deleterious self-aggregation and cytotoxicity [5,6,7]. Thereby, HSA provides ca 60% of the total amyloid inhibitory activity of plasma proteins [5]. Furthermore, the antioxidant and detoxification activities and anti-inflammatory and blood–brain barrier-supporting properties of HSA likely prevent AD progression [8].

Based on the Aβ-buffering role of plasma HSA, removal of plasma Aβ via plasma exchange and replacement with therapeutic HSA were suggested for facilitation of Aβ efflux from the brain of AD patients to plasma [9]. The clinical trials have shown persistent improvements in the memory and language functions of the patients [9,10,11]. An alternative, less invasive approach to the lowering of the free Aβ level in the brain is a pharmacologically guided shift in the equilibrium between HSA and Aβ towards their complex. The latter possibility was recently confirmed by arachidonic/linoleic-acid-induced enhancement of HSA affinity to Aβ [12], which is in line with the observation that dietary linoleic acid reduces brain Aβ deposition in AD mouse models [13]. Meanwhile, many more HSA ligands were reported to prevent HSA-induced suppression of Aβ fibrillation, likely due to stimulation of Aβ release from HSA: tolbutamide [5], warfarin, palmitic acid, and cholesterol [14]. Therefore, searching for HSA ligands with the opposite action is non-trivial.

Serotonin (SRO; 5-hydroxytryptamine, 5-HT) is a HSA ligand closely related to AD pathogenesis (reviewed in [15,16]). It is a phylogenetically conserved monoamine neurotransmitter that performs various physiological functions (behavior regulation, appetite suppression, regulation of energy intake, storage, and expenditure, respiratory drive, hemostasis, etc.) through interaction with multiple 5-HT receptors [17].

Numerous studies have demonstrated degeneration of the serotonin system in AD, a decline in the cortical/serum SRO level, and negative consequences of disruption in serotonergic signaling [15,16]. The serotonergic deficits in AD are more prominent compared with those for other neurotransmitters. Selective serotonin reuptake inhibitors (SSRIs) aimed at increasing the SRO bioavailability at nerve terminals are currently in clinical trials for AD [18]. In the present work, we show in vitro that SRO is the most potent activator of HSA–Aβ interaction reported to date, whereas the structurally similar SRO precursor, L-tryptophan (TRP), is inactive in this sense.

## 2. Results

The HSA–Aβ interaction was studied by the SPR method mainly as described earlier [12]. The recombinant human Aβ_40_ or Aβ_42_ sample was immobilized on the surface of the SPR sensor chip by amine coupling, followed by removal from the surface of the non-covalently bound Aβ molecules by a 2% SDS water solution [12]. A fatty-acid-free HSA sample [19] (5–60 µM) was used as an analyte. The measurements were carried out at 25 °C using a physiologically relevant buffer (20 mM Tris-HCl, 140 mM NaCl, 4.9 mM KCl, 1 mM MgCl_2_, 2.5 mM CaCl_2_, pH 7.4). The resulting SPR sensograms exhibited a concentration-dependent association–dissociation pattern and were well approximated by the one-site binding model (Figure 1). The equilibrium dissociation constants (*K_D_*) for intact HSA complexes with Aβ_40_ and Aβ_42_ were 0.6 × 10^−7^ M and 1.2 × 10^−7^ M, respectively (Table 1), in accord with our previous estimates [12].

Addition of 1 mM TRP to the solution is sufficient to load HSA with TRP, considering the equilibrium association constant of 2.7 × 10^4^ M^−1^ [20]. Nevertheless, 1 mM TRP did not affect HSA’s apparent affinity to Aβ_40_/Aβ_42_ (Figure 1 and Figure 2, Table 1). Instead, it notably changed the kinetics of the HSA–Aβ_42_ interaction, inducing slowdown of the association and dissociation processes (see Figure 3). Meanwhile, addition of 1 mM SRO decreased the apparent *K_D_* value for the HSA–Aβ complex by a factor of 7 and 17, for Aβ_40_ and Aβ_42_, respectively (Figure 1, Table 1), mostly due to a slowdown of the dissociation process (Figure 3). A decrease in the SRO concentration to 10 µM was accompanied by an increase in the apparent *K_D_* up to the values observed in the absence of the ligand (Figure 2, Table 1). Since the equilibrium dissociation constant for the HSA–SRO complex is 1.6 µM [21], 10 µM SRO is enough to saturate HSA with SRO. For this reason, the absence of noticeable effects of 10 µM SRO on HSA’s affinity to Aβ indicates that SRO binding to HSA alone is insufficient for improvement of its affinity to Aβ, thereby suggesting the necessity of direct SRO interaction with Aβ for that. The latter phenomenon is indicated by the efficient inhibition of Aβ_42_ fibrillation by 50–100 µM SRO [22]. Notably, TRP was markedly less effective in this sense. The direct interaction of SRO with both Aβ and HSA implies that the equilibrium and kinetic dissociation/association constants determined by SPR spectroscopy for HSA–Aβ interaction in the presence of SRO(TRP) represent the apparent values, effectively describing the complex network of the multiple chemical equilibria occurring in the system.

The drastic difference between the effects exerted by SRO and TRP on the HSA–Aβ interaction may partly arise due to the structural differences between HSA complexes with SRO/TRP. The SRO/TRP-induced changes in the quaternary structure of HSA were explored by crosslinking with glutaraldehyde at 37 °C, followed by SDS-PAGE analysis (Figure 4A). Intact HSA is mostly monomeric (66%) with minor contributions of dimeric/trimeric and high-molecular-weight forms (about 23% and 12%, respectively), consistent with the previous observations [23]. While addition of 1 mM TRP did not affect the quaternary structure of HSA, addition of 1 mM SRO was accompanied by an approximately 2-fold increase in the content of high-molecular-weight forms at the expense of the monomer. Dynamic light scattering spectroscopy confirmed that SRO binding to HSA induces more prominent accumulation of its multimeric forms, compared with the effects induced by TRP (Figure 4B). Overall, SRO binding caused more distinct changes in the quaternary structure of HSA, relative to the TRP-induced effects.

Since tertiary structures of HSA complexes with SRO/TRP have not been reported to date, we predicted the location of a SRO/TRP-binding pocket in HSA using the DoGSiteScorer algorithm [24]. According to these predictions, the binding site is located in the region between the IB site, ‘IIA: Drug site 1′, and ‘IIIA: Drug site 2′ [25] (Figure 5).

The binding pocket is predominantly apolar, but contains charged residues and clusters of polar residues (Tyr174 of domain I; Ser216, Gln245, Ser311, Tyr365, Ser366 of domain II; Cys472, Tyr476, Ser478 of domain III): Y174, E177, S216, K219, Q220, L222, K223, F235, W238, A239, R242, L243, Q245, R246, F247, L258, L262, H266, R281, L284, A285, I288, L299, K310, S311, H312, I314, A315, E316, E318, D320, P363, Y365, S366, V367, V368, K468, P471, C472, D475, Y476, S478, V479. Molecular docking of HSA with SRO/TRP using AutoDock Vina software [26] predicted the difference in their orientation within the binding pocket (Figure 5; see Supplementary PDB files in Appendix A). Analysis of interactions between HSA and the bound ligands using PLIP software [27] revealed specific hydrogen bonds and π–cation interactions (Table 2, Figure 5B,C). Overall, although the molecular modeling predicted the same binding HSA pocket for SRO/TRP, noticeable differences in structural peculiarities of microenvironments of the bound SRO/TRP molecules were also detected, which may have led to the differences in the HSA structural rearrangements in response to the binding of these ligands.

To explore the conformational flexibility of the established SRO/TRP-binding pocket of HSA, we conducted a multiparametric evaluation of the protein predisposition for intrinsic disorder using a set of commonly utilized per-residue disorder predictors. In these analyses, a threshold of 0.5 is typically used to identify disordered residues and regions in a query protein. Residues with the disorder scores (DS) 0.25 ≤ DS < 0.5 are considered to be moderately disordered, whereas residues with 0.15 ≤ DS < 0.25 are flexible. The results of this analysis are shown in Figure 6, which clearly illustrate that most of the residues involved in the binding to TRP or SRO are characterized by high conformational flexibility. In fact, with a very few exceptions, all such residues are located within regions that are predicted to be flexible/disordered by at least six predictors. Importantly, the several residues are predicted to have a propensity for ordering. This observation suggests that the interplay between intrinsic (amino acid sequence-based) structural flexibility and order propensity is important to accommodating TRP/SRO in the binding site, similarly to the target recognition principle established for S100 proteins [28]. Overall, the disorder propensity predictions suggest that SRO/TRP binding to HSA is not of the “lock-and-key” type but is likely to follow the “induced fit” mechanism.

## 3. Discussion

The literature data on HSA interaction with monomeric Aβ are contradictory [7,35,36,37,38]. While some studies did not detect HSA interaction with monomeric Aβ [35,36], other studies confirmed it [7,37,38], but gave affinity estimates differing from those reported here and in our previous work [12]. These contradictions seem to arise due to the high propensity of Aβ for multimerization, leading to sensitivity of the experiments to many factors, including the source of Aβ samples (recombinant or synthetic), the Aβ pretreatment procedures and solvent conditions (confirmed in [12]), and the methods used for the analysis. To ensure the monomeric state of Aβ in our experiments, we used the previously validated approach based on Aβ immobilization on the surface of a SPR chip by amine coupling, followed by removal of the non-covalently bound Aβ molecules with SDS [12].

Previously, we showed the ability of linoleic and arachidonic acids to favor interaction between HSA and monomeric Aβ_40_/Aβ_42_ with a 3-fold decrease in the *K_D_* value [12]. The data on SRO’s effect on the HSA–Aβ equilibrium presented here illustrate the much higher potential for modulation of this equilibrium by low-molecular-weight HSA ligands with up to a 17-fold decrease in the *K_D_* value. Such a dramatic change in HSA’s affinity for Aβ could be achieved via the following mechanisms or their combination: (I) an allosteric mechanism (ligand binding to HSA, altering its affinity for Aβ); or (II) ligand binding to Aβ, affecting its interaction with HSA. In the case of mechanism I alone, the SRO-induced changes in HSA‘s affinity for Aβ would be noticeable at SRO concentrations above 1.6 µM (the *K_D_* value for the HSA–SRO complex [21]).

The absence of a marked influence of 10 µM SRO on the HSA–Aβ interaction points out the contribution of mechanism II. This conclusion is in line with the data on the potent inhibition of Aβ_42_ fibrillation by 50–100 µM SRO [22], which corresponds to the SRO level ensuring notable changes in HSA’s affinity for Aβ. Despite the similar structures of SRO and TRP, the latter did not affect HSA’s effective affinity for Aβ but slowed their association and dissociation. The drastic difference between SRO/TRP actions on the HSA–Aβ equilibrium can be rationalized within the same mechanisms I and II. The notable difference in quaternary structures of HSA in complex with SRO/TRP and distinct structural differences between HSA–SRO/TRP complexes predicted by molecular docking evidence that SRO binding and TRP binding differently alter HSA’s structure. Meanwhile, the much less efficient inhibition of Aβ_42_ fibrillation by TRP, compared with SRO [22], indicates the possibility of the differences between SRO and TRP in their interaction with monomeric Aβ.

The revealed influence of SRO on the HSA–Aβ interaction could be relevant to AD progression, considering the disintegration of the serotonergic system during AD, accompanied by a decrease in the SRO level in biofluids [15,16,39]. The SRO concentration in plasma is 0.28–1.7 µM [40], but in the synaptic cleft it may reach 6 mM and 55 nM after SRO diffuses into the extracellular compartment [41]. Hence, the peak SRO level in the central nervous system is sufficient for modulation of the HSA–Aβ interaction. Therefore, the decrease in the Aβ level in interstitial brain fluid observed in response to either administration of SSRIs or direct infusion of SRO into the hippocampus in a mouse model of AD [42] could be partly due to the facilitated trapping of Aβ by HSA. Chronic treatment with a SSRI caused a 50% reduction in plaque load in mouse brain, and the same effect has been confirmed for cognitively normal elderly participants chronically exposed to antidepressant drugs [42]. Thus, the principal finding of this work provides an alternative explanation to the available literature data on the SRO-induced reduction in the Aβ level in interstitial brain fluid and the respective decrease in brain amyloid load, thereby highlighting the importance of SRO for proper buffering of Aβ by HSA.

Since chemical structures of agents in clinical studies in AD, including SSRIs aimed at increasing the SRO level at nerve terminals [18,43], indicate their ability to interact with HSA or/and Aβ, it would be reasonable to complement their trials by tests of the ability of the drug candidates to affect the HSA–Aβ interaction.

## 4. Materials and Methods

### 4.1. Materials

A fatty-acid-free HSA sample prepared under non-denaturing conditions (#126654) [19] was purchased from Merck (Darmstadt, Germany). Ultra-pure recombinant human Aβ_42_ produced in *E. coli* (#A-1163-1) was obtained from rPeptide (Watkinsville, GA, USA). Usp2-cc was prepared mainly as described in [44]. The SRO (#B21263) was from Alfa Aesar (Kandel, Germany). The TRP (#A1645) and HEPES were from AppliChem (Darmstadt, Germany). Ultra-grade Tris and 2-ME were purchased from VWR Life Science AMRESCO (Vienna, Austria). The sodium chloride, potassium chloride, urea, and SDS were from Panreac Applichem (Darmstadt, Germany). The calcium chloride and magnesium chloride were from Fluka (Charlotte, NC, USA). Ethylenediaminetetraacetic acid (EDTA), grade II glutaraldehyde, and acetonitrile were from Merck (Darmstadt, Germany). NH_4_OH and acetic acid were purchased from Component-reaktiv (Moscow, Russia) and Labteh (Moscow, Russia), respectively. Trifluoroacetic acid (TFA) was bought from Fisher Sci (Waltham, MA, USA). Dimethyl sulfoxide (DMSO) was purchased from Helicon (Moscow, Russia). Ethanolamine and Profinity^TM^ IMAC Resin were obtained from Bio-Rad Laboratories (Hercules, CA, USA). A Jupiter C18 column was purchased from Phenomenex^®^ (Torrance, CA, USA).

Protein concentrations were measured spectrophotometrically using molar extinction coefficients at 280 nm calculated according to [45]: 34,445 M^−1^cm^−1^ for HSA and 1490 M^−1^cm^−1^ for Aβ_40_/Aβ_42_.

### 4.2. Expression and Purification of Recombinant Human Aβ_40_

Human Aβ_40_ was expressed in *E. coli* and purified as described earlier [12] with some modifications. The 2-Mercaptoethanol (2-ME) concentration in the buffers was 5 mM. The 6×His-ubiquitin-Aβ_40_ fusion protein was dialyzed twice against 50 mM Tris-HCl and 5 mM 2-ME (pH 8.6) buffers containing 6 M and 4 M urea, respectively. Aβ_40_ was excised from the fusion protein via incubation with Usp2-cc (50-fold molar excess of the fusion over the enzyme) at room temperature for 3 h. Urea was added to the solution up to 8 M. The hydrolysate was passed through a Profinity IMAC column equilibrated with 50 mM Tris-HCl, 5 mM 2-ME, and 8 M urea (pH 8.0–8.5) to remove His-tagged ubiquitin and Usp2-cc. Aβ_40_ was purified by chromatography on an HPLC Jupiter C18 column (buffer A: 20 mM NH_4_^+^, 6 mM acetic acid, pH 9.6; buffer B: 70% acetonitrile; gradient steps: loading of the sample in 15% buffer B, flushing by 25% buffer B for 10 min, linear gradient from 28% to 45% buffer B for 80 min). Precise chain cleavage by Usp2-cc was confirmed by electrospray ionization mass spectrometry (LCMS-2010EV, Shimadzu, Kyoto, Japan). The purified Aβ_40_ samples were freeze-dried and stored at −70 °C.

### 4.3. Pretreatment of Aβ Samples for SPR Experiments

Recombinant human Aβ_40_/Aβ_42_ samples were pretreated for SPR measurements mainly as described in [46]. The freeze-dried Aβ sample was dissolved in neat TFA at a concentration of 0.5–1 mg/mL. The solution was sonicated for 30 s. TFA was evaporated using Eppendorf Concentrator plus. The dried Aβ sample was dissolved in DMSO at a concentration of 2 mg/mL, and stored at −20 °C.

### 4.4. Surface Plasmon Resonance Studies

SPR measurements were performed at 25 °C using a Bio-Rad ProteOn™ XPR36 instrument mainly as previously described [12]. Ligand (57 μg/mL recombinant human Aβ_40_/Aβ_42_ in 10 mM sodium acetate, pH 4.5 buffer) was immobilized on a ProteOn GLH sensor chip surface (up to 7000–9000 RUs) by amine coupling. The remaining activated amine groups on the chip surface were blocked by 1 M ethanolamine solution. The non-covalently bound Aβ_40_/Aβ_42_ molecules were washed from the chip surface with 2% SDS water solution until stabilization of the signal [12]. The monomeric state of Aβ was previously verified by use of a polyclonal antibody selective to Aβ oligomers (A11) or an anti-Aβ_20_ monoclonal antibody (7N22) as the analyte [12]; the immobilized Aβ sample did not reveal changes in the SPR signal upon application of A11 but interacted with 7N22. Analyte (5–60 μM HSA) in the running buffer (20 mM Tris-HCl, 140 mM NaCl, 4.9 mM KCl, 1 mM MgCl_2_, 2.5 mM CaCl_2_, pH 7.4) was passed over the chip at a rate of 30 μL/min for 300 s, followed by flushing the chip with the running buffer for 2400 s. The sensor chip surface was regenerated by passage of 10 mM EDTA pH 7.9 solution containing 2% SDS. The influence of HSA ligands on its interaction with Aβ_40_/Aβ_42_ was studied in the presence of TRP/SRO (10 µM, 100 µM, or 1 mM). The double-referenced SPR sensograms were globally fitted according to the one-site binding model using Bio-Rad ProteOn Manager™ v.3.1 software. The effective equilibrium (*K_D_*) and kinetic (*k_d_* and *k_a_*) dissociation/association constants were evaluated for each HSA concentration, followed by averaging of the resulting values (standard deviations are indicated).

### 4.5. Chemical Crosslinking of Proteins

Crosslinking of HSA (36 μM) with 0.02% glutaraldehyde was performed in 20 mM HEPES-KOH, 140 mM NaCl, 4.9 mM KCl, 1 mM MgCl_2_, 2.5 mM CaCl_2_, pH 7.4 buffer in the absence/presence of 1 mM SRO/TRP at 37 °C for 1 h, mainly as described in [23].

### 4.6. Dynamic Light Scattering Studies

DLS measurements of HSA (36 μM) solutions in 20 mM HEPES-KOH, 140 mM NaCl, 4.9 mM KCl, 1 mM MgCl_2_, 2.5 mM CaCl_2_, pH 7.4 buffer in the absence/presence of 1 mM SRO/TRP were performed at 25.0 °C using a Zetasizer Nano ZS spectrometer (Malvern Instruments Ltd., Malvern, UK). The back-scattered light from a 4 mW He/Ne laser (632.8 nm) was collected at an angle of 173°. The acquisition time for a single autocorrelation function was 45 s. The resulting autocorrelation functions are averaged values from 20 measurements. The volume–size distributions were calculated using the following parameters: a refractive index of 1.330 and a viscosity value of 0.8882 cP.

### 4.7. Prediction of Ligand-Binding Sites in HSA

The X-ray structure of HSA was obtained from Protein Data Bank [47] entry 1UOR (chain A). Water molecules were removed from the structure, while hydrogen atoms were added, using AutoDockTools software (http://autodock.scripps.edu/resources/adt/index_html, accessed on 28 April 2021). The three-dimensional structures of SRO and TRP were obtained from the PubChem database (https://www.ncbi.nlm.nih.gov/pccompound, accessed on 28 April 2021) and converted into PDB format using PyMOL v.1.6.9.0 software (https://pymol.org/2/, accessed on 28 April 2021). SRO/TRP-binding pockets of HSA were identified using the DoGSiteScorer algorithm [24] implemented in the ProteinsPlus service (https://proteins.plus/, accessed on 28 April 2021). AutoDock Vina (http://vina.scripps.edu/index.html, accessed on 28 April 2021) [26] was used for molecular docking. The resulting model HSA–SRO/TRP complexes were visualized using PyMOL v.1.6.9.0. Characterization of the protein–ligand interactions was performed by the PLIP service (https://plip-tool.biotec.tu-dresden.de/plip-web/plip/index, accessed on 28 April 2021) [27] using numbering of the residues according to UniProt entry P02768.

### 4.8. Evaluation of the Per-Residue Intrinsic Disorder Predisposition of HSA

The intrinsic disorder predisposition of HSA was analyzed by commonly used per-residue predictors, including PONDR^®^ VLXT [29], PONDR^®^ VSL2 [31], and PONDR^®^ VL3 [30], which are available on the PONDR site (http://www.pondr.com, accessed on 28 April 2021), PONDR^®^ FIT, which is a meta-predictor that incorporates predictions from several different sources [32], and the IUPred computational platform that allows for identification of either short or long regions of intrinsic disorder, IUPred Long and IUPred_Short [33]. We utilized the DiSpi web crawler, which was designed for the rapid prediction and comparison of protein disorder profiles. It aggregates the results from a number of well-known disorder predictors: PONDR^®^ VLXT [29], PONDR^®^ VL3 [30], PONDR^®^ VLS2B [31], PONDR^®^ FIT [32], IUPred2_Short, and IUPred2_Long [33,34]. The outputs of the disorder predictors are represented as real numbers between 1 (ideal prediction of disorder) and 0 (ideal prediction of order). A threshold of ≥0.5 was used to identify disordered residues and regions in HSA. Residues with the disorder scores (DS) 0.25 ≤ DS < 0.5 are considered to be moderately disordered, whereas residues with 0.15 ≤ DS < 0.25 are flexible.

## 5. Conclusions

The presented example of ligand-induced facilitation of HSA interaction with monomeric Aβ demonstrates the high potential for pharmacological modulation of this interaction aimed at decreasing the free Aβ concentration, leading to the inhibition of malicious Aβ deposits. The revealed effect seems to involve ligand binding to both participants of the reaction. The data deepen our understanding of the SRO-mediated effects reported in animal and clinical studies.

## Figures and Tables

**Figure 1 ijms-22-05896-f001:**
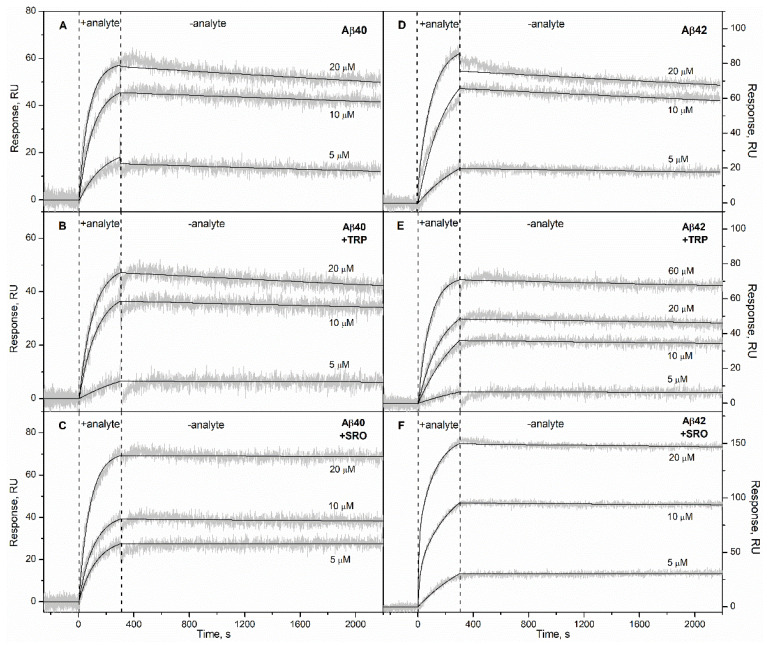
Ligand-dependent interaction of HSA with monomeric Aβ_40_/Aβ_42_ at 25 °C monitored by SPR spectroscopy (20 mM Tris-HCl, 140 mM NaCl, 4.9 mM KCl, 1 mM MgCl_2_, 2.5 mM CaCl_2_, pH 7.4). Aβ_40_/Aβ_42_ was immobilized on the sensor chip’s surface by amine coupling. HSA concentrations are indicated nearby the sensograms for its interaction with Aβ_40_ (panels **A**–**C**) or Aβ_42_ (**D**–**F**) in the absence (**A**,**D**) or presence of 1 mM TRP (**B**,**E**) or 1 mM SRO (**C**,**F**). The grey curves are experimental, while the black curves are theoretical, calculated according to the one-site binding model (see Table 1 for the fitting parameters).

**Figure 2 ijms-22-05896-f002:**
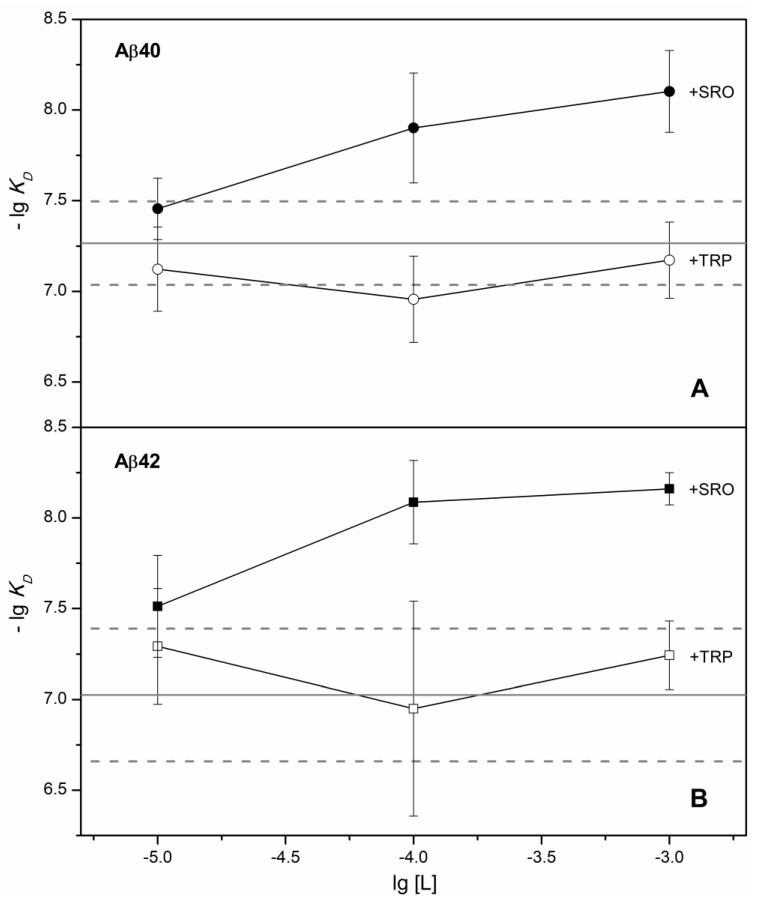
Influence of SRO/TRP (denoted as ‘L’) on the effective equilibrium dissociation constant, *K_D_*, for the HSA complex with monomeric Aβ_40_ (panel **A**) or Aβ_42_ (**B**), according to SPR data (see Table 1). The average-lg *K_D_* values for the HSA–Aβ_40_/Aβ_42_ complexes in the absence of the ligands are marked by solid gray lines, and their boundary values considering standard deviations are indicated by dashed gray lines.

**Figure 3 ijms-22-05896-f003:**
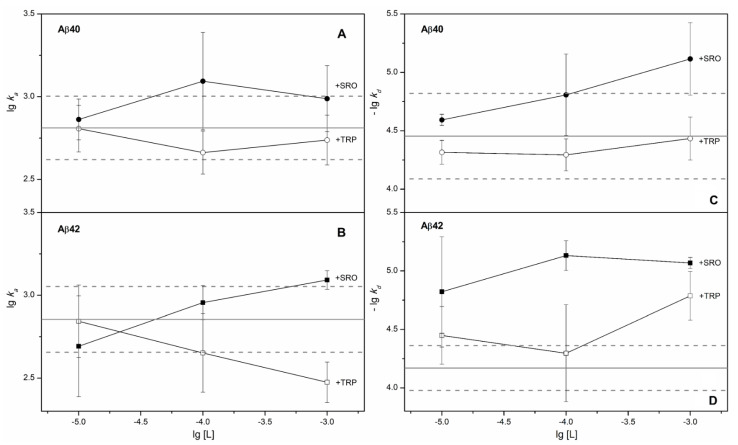
Influence of SRO/TRP on the association and dissociation processes. Effect of the SRO/TRP on the effective kinetic association constant, *k_a_*, for HSA interaction with monomeric Aβ40 (panel **A**) or Aβ42 (**B**), according to SPR data (see Table 1). The average lg *k_a_* values for the HSA–Aβ40/42 complexes in the absence of the ligands are marked by solid gray lines, and their boundary values considering standard deviations are indicated by dashed gray lines. Influence of SRO/TRP on the effective kinetic dissociation constant, *k_d_*, for the HSA complex with monomeric Aβ40 (panel **C**) or Aβ42 (**D**), according to SPR data (see Table 1). The average-lg *k_d_* values for the HSA–Aβ40/42 complexes in the absence of the ligands are marked by solid gray lines, and their boundary values considering standard deviations are indicated by dashed gray lines.

**Figure 4 ijms-22-05896-f004:**
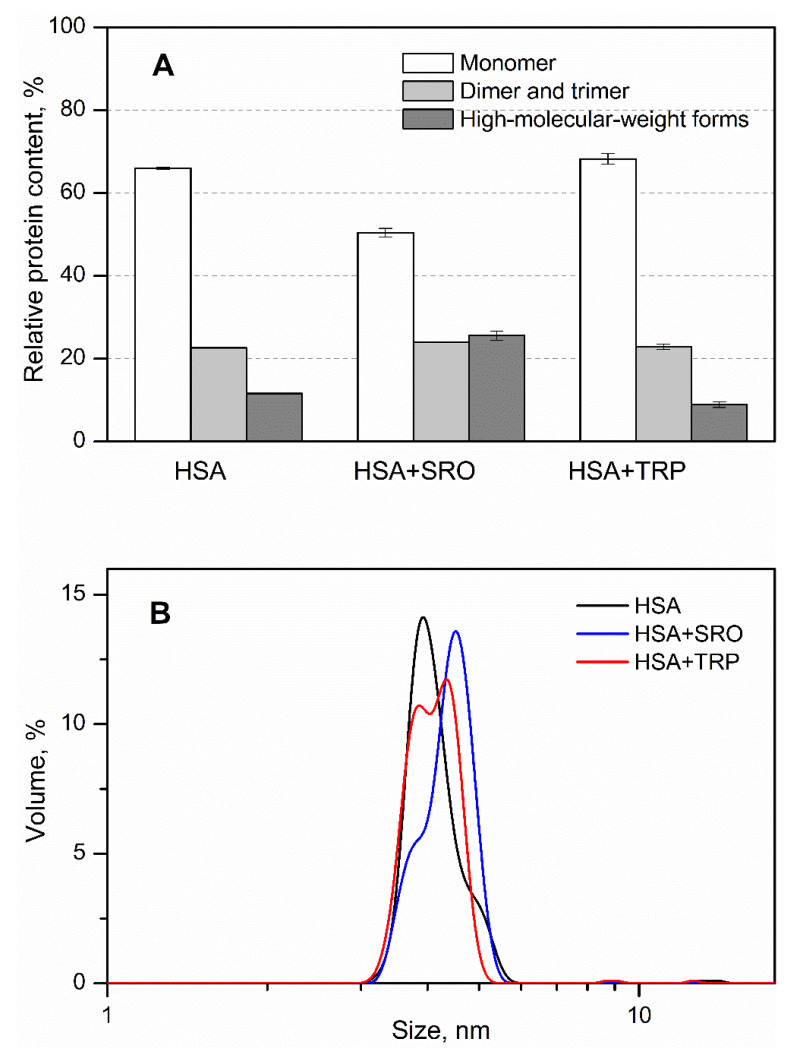
Influence of SRO/TRP (1 mM) on the distribution of HSA (36 μM) over multimeric forms, as judged from glutaraldehyde crosslinking experiments at 37 °C (panel **A**) and DLS at 25 °C (**B**) (20 mM HEPES-KOH, 140 mM NaCl, 4.9 mM KCl, 1 mM MgCl_2_, 2.5 mM CaCl_2_, pH 7.4). The standard deviations are indicated.

**Figure 5 ijms-22-05896-f005:**
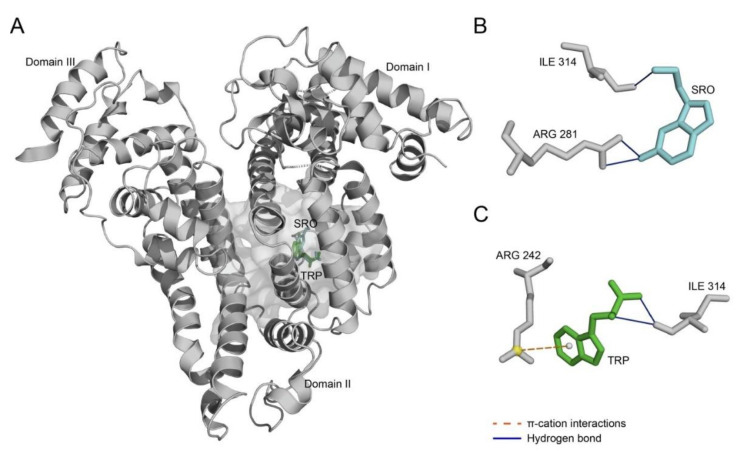
Overlay of model tertiary structures of HSA-SRO/TRP complexes predicted using AutoDock Vina software [26], visualized with PyMOL (panel **A**). The ligand-binding pocket (grey) is represented as a surface. Representation of HSA residues interacting with SRO (cyan) (**B**) and TRP (green) (**C**).

**Figure 6 ijms-22-05896-f006:**
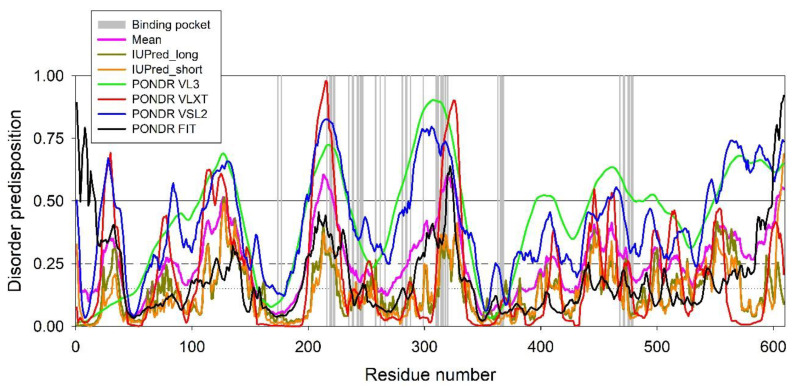
Bioinformatics analysis of the disorder predisposition of the full-length HSA (UniProt ID: P02768), including the signal peptide (residues 1–18) and the pro-peptide (residues 19–24). Intrinsic disorder profiles were generated based on the commonly used disorder predictors PONDR^®^ VLXT [29], PONDR^®^ VL3 [30], PONDR^®^ VLS2B [31], PONDR^®^ FIT [32], IUPred2 (Short), and IUPred2 (Long) [33,34]. Mean disorder scores (DS) calculated by averaging the outputs of individual predictors are shown as well. The outputs of the disorder predictors are represented as real numbers between 1 (ideal prediction of disorder) and 0 (ideal prediction of order). A threshold of 0.5 was used to identify disordered residues and regions in HSA. Residues with 0.25 ≤ DS < 0.5 are considered as moderately disordered, whereas residues with 0.15 ≤ DS < 0.25 are flexible. The position of residues involved in TRP/SRO binding is shown by gray vertical bars.

**Table 1 ijms-22-05896-t001:** Parameters of the one-site binding model describing the SPR sensograms for the interaction between HSA and monomeric Aβ_40_/Aβ_42_ in the absence/presence of SRO/TRP.

HSA Ligand	[HSA Ligand]	Aβ_40_			Aβ_42_		
		*k_a_* × 10^−3^, M^−1^s^−1^	*k_d_* × 10^4^, s^−1^	*K_D_* × 10^7^, M	*k_a_* × 10^−3^, M^−1^s^−1^	*k_d_* × 10^4^, s^−1^	*K_D_* × 10^7^, M
**-**	-	0.7 ± 0.3	0.4 ± 0.3	0.6 ± 0.3	0.8 ± 0.3	0.7 ± 0.3	1.2 ± 0.7
SRO	10 µM	0.7 ± 0.2	0.26 ± 0.03	0.4 ± 0.2	0.6 ± 0.4	0.20 ± 0.14	0.4 ± 0.2
	100 µM	1.5 ± 1.2	0.19 ± 0.12	0.15 ± 0.13	0.9 ± 0.2	0.08 ± 0.02	0.09 ± 0.04
	1 mM	1.0 ± 0.5	0.09 ± 0.05	0.09 ± 0.05	1.2 ± 0.2	0.086 ± 0.009	0.070 ± 0.015
TRP	10 µM	0.7 ± 0.2	0.49 ± 0.12	0.8 ± 0.4	0.8 ± 0.4	0.4 ± 0.2	0.6 ± 0.4
	100 µM	0.47 ± 0.15	0.53 ± 0.17	1.2 ± 0.6	0.5 ± 0.3	0.7 ± 0.6	1.8 ± 1.8
	1 mM	0.6 ± 0.2	0.4 ± 0.2	0.7 ± 0.4	0.29 ± 0.07	0.18 ± 0.07	0.6 ± 0.3

**Table 2 ijms-22-05896-t002:** The interactions between SRO/TRP and HSA within the model of their complex shown in Figure 5. The distance between donor and acceptor atoms is indicated for hydrogen bonds.

Ligand	Index	Residue #	Residue	Distance, Å
SRO	**Hydrogen bonds**			
	1	281	Arg	3.16
	2	281	Arg	3.18
	3	314	Ile	2.22
TRP	**Hydrogen bonds**			
	1	314	Ile	2.99
	2	314	Ile	3.21
	**π–cation interactions**			
	1	242	Arg	5.57

## Data Availability

Not applicable.

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
