# Peer review of "Serotonin Promotes Serum Albumin Interaction with the Monomeric Amyloid β Peptide"

_ijms, 2021, doi:10.3390/ijms22115896_

Round 1
Reviewer 1 Report
In the manuscript by Litus demonstrated the physiological relevant binding activity of HSA to abeta(1-40)/(1-42) in the presence of serotonin. The loss of serotonin, depression, and the promotion of AD severity is well known fact, and the study will bring the new hypothesis for the serotonin/Ad hypothesis based on the molecular interaction. This discovery is physiologically relevant, and the manuscript is acceptable, only a fer points should be appropriately addressed.
- Not a good use of terminology in the article title.
The title of this article is “Serotonin Promotes Serum Albumin Interaction with Monomeric Amyloid β Peptide”. However, in the authors’ main experiment, both Ab(1-40) and Ab(1-42) were immobilized onto the SPR sensor chips, thus the presence of the isolated monomer was not completely guaranteed. In addition, as shown in the following references in the next issue, albumin interaction against either monomeric or oligomeric Ab peptides are still controversial. Thus, the referee recommends using the other terminology, such like “non-fibrillated”.
- According to the previous issue, the molecular specificity of the albumin (both bovine and human) to monomeric or oligomeric Ab peptides are still controversial. The authors may better adding a shore discussion section for explaining this point with referring the several important references;
Kuo, et al., Biochem Biophys Res Commun, 2000 Feb 24;268(3):750-6. doi: 10.1006/bbrc.2000.2222.
“Amyloid-beta peptides interact with plasma proteins and erythrocytes: implications for their quantitation in plasma.”
Rozga et al.. Biochem Biophys Res Commun. 2007 Dec 21;364(3):714-8. doi: 10.1016/j.bbrc.2007.10.080. Epub 2007 Oct 23.
“The binding constant for amyloid Abeta40 peptide interaction with human serum albumin.”
Milojevuc et al., J Alzheimers Dis. 2014;38(4):753-65. doi: 10.3233/JAD-131169.
“In vitro amyloid-β binding and inhibition of amyloid-β self-association by therapeutic albumin.”
- Absence of the material and method section for sample preparation and the protocol for the pre-treatment of Ab(1-42). Especially, if the authors want to claim for “monomeric” Ab(1-42), the detailed protocol for keeping Ab(1-42) as monomeric form (i.e. NaOH pre-treatment or the condition of HFIP treatment) should be described. Since many experiments using Ab(1-42) seems less reproducible because of the lack of these experimental conditions, such the information is especially valuable for the researchers in the field.
- The importance of the figure 5 in the context is unclear. The presented data of HSA/SRO binding does not indicate any importance of the presence of intrinsically disordered region in HSA. Thus, many readers may confuse the logical flow of the authors working hypothesis. However, recently, a hypothesis called “molecular shield hypothesis” has been emerged, in which IDP can efficiently inhibit amyloid formation (Ikeda et al). It is better the authors may discuss about this point more clearly.
Ikeda et al., Sci Rep. 2020 Jul 23;10(1):12334. doi: 10.1038/s41598-020-69129-1.
“Presence of intrinsically disordered proteins can inhibit the nucleation phase of amyloid fibril formation of Aβ(1-42) in amino acid sequence independent manner.”
Author Response
In the manuscript by Litus demonstrated the physiological relevant binding activity of HSA to abeta(1-40)/(1-42) in the presence of serotonin. The loss of serotonin, depression, and the promotion of AD severity is well known fact, and the study will bring the new hypothesis for the serotonin/Ad hypothesis based on the molecular interaction. This discovery is physiologically relevant, and the manuscript is acceptable, only a fer points should be appropriately addressed.
- Not a good use of terminology in the article title.
The title of this article is “Serotonin Promotes Serum Albumin Interaction with Monomeric Amyloid β Peptide”. However, in the authors’ main experiment, both Ab(1-40) and Ab(1-42) were immobilized onto the SPR sensor chips, thus the presence of the isolated monomer was not completely guaranteed. In addition, as shown in the following references in the next issue, albumin interaction against either monomeric or oligomeric Ab peptides are still controversial. Thus, the referee recommends using the other terminology, such like “non-fibrillated”.
ANSWER: The monomeric state of Ab(1-40)/(1-42) is our SPR experiments has been established in our previous work (https://doi.org/10.1016/j.bbrc.2019.01.081) by use of polyclonal antibody selective to Ab oligomers (A11) or anti-Ab20 monoclonal antibody (7N22) as the analyte: the immobilized Ab sample did not reveal changes in SPR signal upon application of A11, but interacted with 7N22. We have added the corresponding explanations to the “Surface plasmon resonance studies” section of the “Materials and methods”. This is the only case of the SPR experiment with proven monomeric state of Ab reported in the literature, to the best of our knowledge. Therefore, we believe that it is necessary to emphasize that the present work is aimed at studies of HSA interaction with monomeric Ab.
- According to the previous issue, the molecular specificity of the albumin (both bovine and human) to monomeric or oligomeric Ab peptides are still controversial. The authors may better adding a shore discussion section for explaining this point with referring the several important references;
Kuo, et al., Biochem Biophys Res Commun, 2000 Feb 24;268(3):750-6. doi: 10.1006/bbrc.2000.2222.
“Amyloid-beta peptides interact with plasma proteins and erythrocytes: implications for their quantitation in plasma.”
Rozga et al.. Biochem Biophys Res Commun. 2007 Dec 21;364(3):714-8. doi: 10.1016/j.bbrc.2007.10.080. Epub 2007 Oct 23.
“The binding constant for amyloid Abeta40 peptide interaction with human serum albumin.”
Milojevuc et al., J Alzheimers Dis. 2014;38(4):753-65. doi: 10.3233/JAD-131169.
“In vitro amyloid-β binding and inhibition of amyloid-β self-association by therapeutic albumin.”
ANSWER: We have added the following explanations to the Discussion section:
“The literature data on HSA interaction with monomeric Aβ are contradictory [1-5]. While some studies did not detect HSA interaction with monomeric Aβ [1, 2], other studies confirmed it [3-5], but gave the affinity estimates differing from those reported here and in our previous work [6]. These contradictions seem to arise due to high propensity of Aβ to multimerization, leading to sensitivity of the experiments to many factors, including source of Aβ samples (recombinant or synthetic), Aβ pretreatment procedures and solvent conditions (confirmed in ref. [6]), and methods used for the analysis. To ensure monomeric state of Aβ in our experiments, we have used the previously validated approach based on Aβ immobilization on the surface of SPR chip by amine coupling, followed by removal of the non-covalently bound Aβ molecules with SDS [6].”
[1] J. Milojevic, M. Costa, A.M. Ortiz, J.I. Jorquera, G. Melacini, In vitro amyloid-β binding and inhibition of amyloid-β self-association by therapeutic albumin, J Alzheimers Dis 38 (2014) 753-765.
[2] J. Milojevic, A. Raditsis, G. Melacini, Human serum albumin inhibits Abeta fibrillization through a "monomer-competitor" mechanism, Biophys J 97 (2009) 2585-2594.
[3] M. Rózga, M. Kłoniecki, A. Jabłonowska, M. Dadlez, W. Bal, The binding constant for amyloid Aβ40 peptide interaction with human serum albumin, Biochemical and Biophysical Research Communications 364 (2007) 714-718.
[4] C. Wang, F. Cheng, L. Xu, L. Jia, HSA targets multiple Aβ42 species and inhibits the seeding-mediated aggregation and cytotoxicity of Aβ42 aggregates, RSC Advances 6 (2016) 71165-71175.
[5] M. Costa, A.M. Ortiz, J.I. Jorquera, Therapeutic albumin binding to remove amyloid-β, J Alzheimers Dis 29 (2012) 159-170.
[6] E.A. Litus, A.S. Kazakov, A.S. Sokolov, E.L. Nemashkalova, E.I. Galushko, U.F. Dzhus, V.V. Marchenkov, O.V. Galzitskaya, E.A. Permyakov, S.E. Permyakov, The binding of monomeric amyloid β peptide to serum albumin is affected by major plasma unsaturated fatty acids, Biochemical and Biophysical Research Communications 510 (2019) 248-253.
- Absence of the material and method section for sample preparation and the protocol for the pre-treatment of Ab(1-42). Especially, if the authors want to claim for “monomeric” Ab(1-42), the detailed protocol for keeping Ab(1-42) as monomeric form (i.e. NaOH pre-treatment or the condition of HFIP treatment) should be described. Since many experiments using Ab(1-42) seems less reproducible because of the lack of these experimental conditions, such the information is especially valuable for the researchers in the field.
ANSWER: We have added the corresponding section (“Pretreatment of Aβ samples for SPR experiments”) to “Materials and methods”.
- The importance of the figure 5 in the context is unclear. The presented data of HSA/SRO binding does not indicate any importance of the presence of intrinsically disordered region in HSA. Thus, many readers may confuse the logical flow of the authors working hypothesis. However, recently, a hypothesis called “molecular shield hypothesis” has been emerged, in which IDP can efficiently inhibit amyloid formation (Ikeda et al). It is better the authors may discuss about this point more clearly.
Ikeda et al., Sci Rep. 2020 Jul 23;10(1):12334. doi: 10.1038/s41598-020-69129-1.
“Presence of intrinsically disordered proteins can inhibit the nucleation phase of amyloid fibril formation of Aβ(1-42) in amino acid sequence independent manner.”
ANSWER: We respectfully disagree with the statement that importance of the figure 5 in the context is unclear. In our view, Figure 5 delivers a very important message - binding of SRO or TRP to HSA occurs by means of the residues preferentially located within disordered/flexible regions. This clearly indicates the role of conformational flexibility of corresponding regions (reflected in their intrinsic predisposition for intrinsic disorder/structural flexibility) in binding of these ligands and also suggests that the SRO/TRP binding to HSA is not of the "lock-and-key" type but is likely to follow the “induced fit” mechanism. We added the corresponding note to the revised manuscript (see p.6). As far as the “molecular shield hypothesis” is concerned, we do not think that it is applicable in our case, as hypothesis is proposing that the IDP can efficiently inhibit amyloid formation, but we are investigating the role of SRO in modulation of the HSA-Aβ interaction.
Reviewer 2 Report
Binding of amyloid β peptide (Aβ) to human serum albumin (HSA) has been implicated as a natural mechanism that lowers the concentration of free Aβ, thus preventing Aβ self-segregation that counteracts the development and progression of Alzheimer's disease. In the present manuscript, the authors follow the hypothesis that natural ligands, like serotonin and its precursor tryptophan, are able to promote enhanced Aβ binding to HSA, thus contributing to effective removal of free Aβ. The authors used surface plasmon resonance to quantify the influential role of serotonin and tryptophan on Aβ-HSA complex formation. Additionally, they performed crosslinking experiments, which support the assumption that SRO binding induces changes in the quaternary structure of HSA. With molecular docking they demonstrate differences between SRO and TRP complexes with HSA.
My major concerns are:
- In Figure 1, the association kinetics appear to be slightly affected (by a factor of two) and changes in the dissociation kinetics can only be discriminated (if at all) from the fitted curves. I am not sure if the fitting was accurately done. I see large deviation in the residuals of the fitted curves. The large changes in Kd claimed by the authors, are mainly attributed to the changes in koff, but I hardly can see such drastic changes in the transients. Can the authors provide an explanation to this? When using high SRO and TRP concentrations (1mM), there is an inconsistency in the general trend, meaning higher concentrations do not accordingly increase the effect on complex formation, Koff values deviate largely between low and high concentrations of the ligands. I guess high concentrations somehow interfere with the method/signal. The authors should carefully readdress this aspect for an accurate assignment of Kd values and interpretation of the observed effects. One could think of another method to validate the Kd values, such as differential scanning calorimetry, isothermal titration calorimetry, microscale thermophoresis, etc...
- The results and interpretation of the crosslinking experiments are not convincing. The authors should validate the findings using analytical size exclusion chromatography and dynamic light scattering or analytical ultracentrifugation. I think this aspect is important for understanding the mechanism.
- While the docking experiments are an important contribution to the findings, the outcome of the docking provides only minor insights into a possible mechanism of how SRO and TRP may influence Aβ-HSA complex formation. Molecular dynamics simulation would be a more elegant approach to mechanistically assess the effect of SRO/TRP.
Author Response
Research article is very interesting and it will provide important information regarding the therapeutic target for the Alzheimer's disease.
I have only few comments regarding the SPR experimental section.
Did authors tried to study the interaction of serotonin with amyloid beta protein with SPR without including the human serum albumin.
ANSWER: Unfortunately, sensitivity of Bio-Rad ProteOn™ XPR36 surface plasmon resonance spectrometer is insufficient for studies of the direct interaction between such low-molecular-weight compounds as serotonin and Ab40/42
It would be interesting to study other neurotransmitters such as GABA or Acetylcholine and others.
ANSWER: We are grateful to the reviewer for this helpful idea. We are currently focused on the ligands revealed by means of bioinformatic analysis of available databases and literature data. The candidates with proven affinity to HSA and related to Alzheimer’s disease (AD) progression are considered as top priority. In fact, secretion of GABA is associated with Alzheimer’s disease (http://dx.doi.org/10.1038/nm.3639), and some of GABA derivatives bind HSA (https://doi.org/10.1186/s40064-016-2752-x). Therefore, GABA interaction with HSA could be relevant to AD. We’ll consider inclusion of GABA and other neurotransmitters into our list of the candidates to be studied in more detail.
Reviewer 3 Report
Research article is very interesting and it will provide important information regarding the therapeutic target for the Alzheimer's disease.
I have only few comments regarding the SPR experimental section.
Did authors tried to study the interaction of serotonin with amyloid beta protein with SPR with out including the human serum albumin.
It would be interesting to study other neurotransmitters such as GABA or Acetylcholine and others.
Author Response
Binding of amyloid β peptide (Aβ) to human serum albumin (HSA) has been implicated as a natural mechanism that lowers the concentration of free Aβ, thus preventing Aβ self-segregation that counteracts the development and progression of Alzheimer's disease. In the present manuscript, the authors follow the hypothesis that natural ligands, like serotonin and its precursor tryptophan, are able to promote enhanced Aβ binding to HSA, thus contributing to effective removal of free Aβ. The authors used surface plasmon resonance to quantify the influential role of serotonin and tryptophan on Aβ-HSA complex formation. Additionally, they performed crosslinking experiments, which support the assumption that SRO binding induces changes in the quaternary structure of HSA. With molecular docking they demonstrate differences between SRO and TRP complexes with HSA.
My major concerns are:
- In Figure 1, the association kinetics appear to be slightly affected (by a factor of two) and changes in the dissociation kinetics can only be discriminated (if at all) from the fitted curves. I am not sure if the fitting was accurately done. I see large deviation in the residuals of the fitted curves. The large changes in Kd claimed by the authors, are mainly attributed to the changes in koff, but I hardly can see such drastic changes in the transients. Can the authors provide an explanation to this?
ANSWER: To visualize the differences between the dissociation processes more clearly we have extended the time scales in Figure 1 up to 2200 s.
The computational accuracy of the kinetic dissociation/association constants was about 10% for all HSA concentrations.
Higher residuals are observed for some of the fitted curves in the very beginning of the dissociation process, which is typical for SPR experiments.
When using high SRO and TRP concentrations (1mM), there is an inconsistency in the general trend, meaning higher concentrations do not accordingly increase the effect on complex formation, Koff values deviate largely between low and high concentrations of the ligands. I guess high concentrations somehow interfere with the method/signal. The authors should carefully readdress this aspect for an accurate assignment of Kd values and interpretation of the observed effects.
ANSWER: In fact, the values of kinetic dissociation/association constants estimated at various HSA concentrations differ by a factor of 2-4 or less, and lack evident trends. This difference is taken into consideration by averaging of the values determined at various HSA concentrations, as explicitly indicated in the “Materials and methods” chapter. The resulting values and their standard deviations are shown in Figures S1 and S2. All interpretations of the observed effects take into account the standard deviations of the kinetic and equilibrium dissociation/association constants.
One could think of another method to validate the Kd values, such as differential scanning calorimetry, isothermal titration calorimetry, microscale thermophoresis, etc...
ANSWER: We agree that use of alternative approaches to the Kd estimates would strengthen the SPR data. Nevertheless, the aforementioned techniques would require use of Aβ concentrations, which are incompatible with monomeric state of Aβ. In fact, we use here the only validated to date approach to preparation of monomeric Aβ, described in ref. https://doi.org/10.1016/j.bbrc.2019.01.081. This method was verified by polyclonal antibody selective to Aβ oligomers (A11) and anti-Aβ20 monoclonal antibody (7N22) used as the analyte: the immobilized Aβ sample did not reveal changes in SPR signal upon application of A11, but interacted with 7N22. We have added the respective explanations to the “Surface plasmon resonance studies” section of the “Materials and methods” chapter.
- The results and interpretation of the crosslinking experiments are not convincing. The authors should validate the findings using analytical size exclusion chromatography and dynamic light scattering or analytical ultracentrifugation. I think this aspect is important for understanding the mechanism.
ANSWER: Since size exclusion chromatography shifts the equilibria between multimeric forms of a protein during their separation, we have added the dynamic light scattering data to Figure 3. DLS spectroscopy confirmed that SRO binding to HSA induces more prominent accumulation of its multimeric forms, compared to the effects induced by TRP. The corresponding changes have been introduced into the “Abstract”, “Methods and materials”, “Results” and “Discussion” sections.
- While the docking experiments are an important contribution to the findings, the outcome of the docking provides only minor insights into a possible mechanism of how SRO and TRP may influence Aβ-HSA complex formation. Molecular dynamics simulation would be a more elegant approach to mechanistically assess the effect of SRO/TRP.
ANSWER: We agree that molecular dynamics analysis would be helpful in assessment of SRO/TRP impact on HSA structure. Meanwhile, the Aβ-binding site of serum albumin remains unknown. Therefore, the detailed knowledge of structural rearrangements in albumin in response to SRO/TRP binding does not ensure elucidation of the mechanism of SRO/TRP influence on albumin affinity to Aβ.
Round 2
Reviewer 2 Report
The authors have improved the quality of their manuscript and clarified some important points. The new DLS data add to the understanding of the effects on quaternary structure induced by ligand binding on HSA. In addition, by showing extended time scales of the SPR traces, the changes become more apparent. However, I am still wondering, why the dissociation rates depend on the concentration of the ligands? E.g. for SRO kdiss decreases with increasing SRO for both Aβs, while in the case of TRP the kdiss from Aβ40 are mostly unaffected by the concentration. Contrary, for Aβ42 kdiss decreases at 10 µM TRP, then it remains unchanged at 100 µM like without TRP, and decreases again at 1 mM TRP. One would expect that the overall dissociation kinetics, as determined by a rate-limiting step, would not depend on the concentration of the ligand, assuming that no other effects like oligomerization-associated cooperativity or other intermediate states would influence the kinetics. Particularly in view that the Kd values vary by approx. a factor of two in the presence of the ligand, small inaccuracies in the determination of kon and koff may have a high impact on Kd and over-estimation of its value. I think it is worth to critically discuss this point. The supplementary figures S1 and S2 have not been provided; at least I cannot find them in the manuscript.
Author Response
The authors have improved the quality of their manuscript and clarified some important points. The new DLS data add to the understanding of the effects on quaternary structure induced by ligand binding on HSA. In addition, by showing extended time scales of the SPR traces, the changes become more apparent. However, I am still wondering, why the dissociation rates depend on the concentration of the ligands? E.g. for SRO kdiss decreases with increasing SRO for both Aβs, while in the case of TRP the kdiss from Aβ40 are mostly unaffected by the concentration. Contrary, for Aβ42 kdiss decreases at 10 µM TRP, then it remains unchanged at 100 µM like without TRP, and decreases again at 1 mM TRP. One would expect that the overall dissociation kinetics, as determined by a rate-limiting step, would not depend on the concentration of the ligand, assuming that no other effects like oligomerization-associated cooperativity or other intermediate states would influence the kinetics. Particularly in view that the Kd values vary by approx. a factor of two in the presence of the ligand, small inaccuracies in the determination of kon and koff may have a high impact on Kd and over-estimation of its value. I think it is worth to critically discuss this point. The supplementary figures S1 and S2 have not been provided; at least I cannot find them in the manuscript.
REPLY:
Some of the statements of the reviewer are not fully correct, considering accuracy of the experimentally determined kinetic dissociation constants:
- “SRO kdiss decreases with increasing SRO for both Aβs” is valid only for Aβ40, 10 mcM and 1 mM SRO;
- “Aβ42 kdiss decreases at 10 µM TRP” is invalid due to the lack of noticeable effect.
These facts are clearly seen from Figure S2. Since supplementary Figures S2 and S1 likely were lost in the process of the manuscript submission, we have added them into the main document as new Figure 3.
In fact, the HSA-Aβ interaction in the presence of SRO/TRP (‘ligand’) is a complex process, which, in the general case, involves interaction of the ligand with the both receptors (proteins). Therefore, we are dealing with the multiple chemical equilibria, which form a complex network of interactions. To emphasize this fact, we have added the following explanations into the Results section:
“The direct interaction of SRO with both Aβ and HSA implies that the equilibrium and kinetic dissociation/association constants determined by SPR spectroscopy for HSA-Aβ interaction in the presence of SRO(TRP) represent the apparent values, effectively describing the complex network of the multiple chemical equilibria occurring in the system”.
Therefore, the dependence of the apparent equilibrium and kinetic dissociation/association constants upon the ligand concentration is not surprising, and even expected. We expectedly observe tendency to saturation of the parameters at the elevated ligand concentrations.
As for accuracy of the equilibrium dissociation constants (Kds), it has been estimated by averaging of the Kd values determined at various HSA concentrations, as explicitly indicated in the “Materials and methods” chapter. Hence, the errors of the kinetic constants do not directly affect accuracy of the resulting Kd values.